# Silencing CDCA8 Suppresses Hepatocellular Carcinoma Growth and Stemness via Restoration of ATF3 Tumor Suppressor and Inactivation of AKT/β–Catenin Signaling

**DOI:** 10.3390/cancers13051055

**Published:** 2021-03-02

**Authors:** Taewon Jeon, Min Ji Ko, Yu-Ri Seo, Soo-Jung Jung, Daekwan Seo, So-Young Park, Keon Uk Park, Kwang Seok Kim, Mikyung Kim, Ji Hae Seo, In-Chul Park, Min-Ji Kim, Jae-Hoon Bae, Dae-Kyu Song, Chi Heum Cho, Jae-Ho Lee, Yun-Han Lee

**Affiliations:** 1Department of Molecular Medicine, Keimyung University School of Medicine, Daegu 42601, Korea; twjeon36@gmail.com (T.J.); minyko@hanmail.net (M.J.K.); uree512@naver.com (Y.-R.S.); se01022@naver.com (S.-Y.P.); zi5595@naver.com (M.-J.K.); 2Molecular and Cellular Biology Graduate Program, University of Massachusetts, Amherst, MA 01003, USA; 3Department of Anatomy, Keimyung University School of Medicine, Daegu 42601, Korea; soojung4234@naver.com; 4Department of Bioinformatics, Psomagen Inc., Rockville, MD 20850, USA; daekwan.seo@psomagen.com; 5Department of Internal Medicine, Keimyung University School of Medicine, Daegu 42601, Korea; kupark@dsmc.or.kr; 6Division of Radiation Cancer Research, Korea Institute of Radiological & Medical Sciences, Seoul 01812, Korea; kskim@kirams.re.kr (K.S.K.); parkic@kirams.re.kr (I.-C.P.); 7Department of Biochemistry, Keimyung University School of Medicine, Daegu 42601, Korea; cjstk2227@nate.com (M.K.); seojh@kmu.ac.kr (J.H.S.); 8Department of Physiology, Keimyung University School of Medicine, Daegu 42601, Korea; jhbae@gw.kmu.ac.kr (J.-H.B.); dksong@gw.kmu.ac.kr (D.-K.S.); 9Department of Obstetrics and Gynecology, Keimyung University School of Medicine, Daegu 42601, Korea; c0035@dsmc.or.kr

**Keywords:** CDCA8, mitosis, HCC, tumor growth, ATF3, GADD34, stemness, Akt, β-catenin

## Abstract

**Simple Summary:**

Although the overexpression of CDCA8 is frequently observed in hepatocellular carcinoma (HCC) tissues, the functions of CDCA8 during HCC development remain to be clarified. The aim of our study was to investigate if targeting CDCA8 could affect liver tumor phenotypes in vitro and in vivo and to identify underlying molecular mechanisms to exert its therapeutic effect. We found that silencing of CDCA8 by siRNA inhibits the growth of parental cancer cell culture and mice tumors and suppresses stemness of CD133^+^ cancer stem cell population through the common responses of the upregulation of the tumor suppressive ATF3/GADD34 functional pathway and inactivation of the Akt/β–catenin signaling axis. These findings suggest CDCA8 as a novel therapeutic target for both primary HCC treatment and the prevention of metastasis or recurrence providing mode of action performed by a CDCA8 inhibitor.

**Abstract:**

Big data analysis has revealed the upregulation of cell division cycle associated 8 (CDCA8) in human hepatocellular carcinoma (HCC) and its poorer survival outcome. However, the functions of CDCA8 during HCC development remain unknown. Here, we demonstrate in vitro that CDCA8 silencing inhibits HCC cell growth and long-term colony formation and migration through the accumulation of the G2/M phase cell population. Conversely, CDCA8 overexpression increases the ability to undergo long-term colony formation and migration. RNA sequencing and bioinformatic analysis revealed that CDCA8 knockdown led to the same directional regulation in 50 genes (25 down- and 25 upregulated). It was affirmed based on protein levels that CDCA8 silencing downregulates the levels of cyclin B1 and p-cdc2 and explains how it could induce G2/M arrest. The same condition increased the protein levels of tumor-suppressive ATF3 and GADD34 and inactivated AKT/β–catenin signaling, which plays an important role in cell growth and stemness, reflecting a reduction in sphere-forming capacity. Importantly, it was demonstrated that the extent of CDCA8 expression is much greater in CD133^+^ cancer stem cells than in CD133^−^ cancer cells, and that CDCA8 knockdown decreases levels of CD133, p-Akt and β-catenin and increases levels of ATF3 and GADD34 in the CD133^+^ cancer stem cell (CSC) population. These molecular changes led to the inhibition of cell growth and sphere formation in the CD133^+^ cell population. Targeting CDCA8 also effectively suppressed tumor growth in a murine xenograft model, showing consistent molecular alterations in tumors injected with CDCA8siRNA. Taken together, these findings indicate that silencing CDCA8 suppresses HCC growth and stemness via restoring the ATF3 tumor suppressor and inactivating oncogenic AKT/β–catenin signaling, and that targeting CDCA8 may be the next molecular strategy for both primary HCC treatment and the prevention of metastasis or recurrence.

## 1. Introduction

Hepatocellular carcinoma (HCC) was the sixth most frequent malignant tumor and the fourth highest cause of cancer death worldwide in 2018, with about 782,000 deaths and 841,000 new cases [1]. The prevalence of HCC varies geographically [2]. Chronic hepatitis B virus infection is the leading cause of HCC; other factors such as alcohol abuse, obesity, hepatitis C virus infection, and type 2 diabetes have also been implicated. Most HCCs are induced by chronic inflammation caused by viral infections, resulting in continuous hepatic inflammation and regeneration of hepatocytes [3]. Regarding therapeutic efficacy, late diagnosis, the presence of other hepatic complications, and a lack of therapy options are limitations in HCC treatment. Conventional treatment options such as liver transplantation, surgical resection, and thermal ablation are performed on HCC patients in the very early or early stages [4,5]. However, liver transplantation is limited by donor organ availability, and the disadvantage of surgical resection is that it does not eliminate parts of the liver that lack function and are at risk of malignant transformation [6]. Currently, sorafenib, a potent small molecule inhibitor of multiple kinases, is the most recommended prescriptive option for the advanced stage (BCLC stage C) of HCC, with invasive or extrahepatic tumor tissues. However, this standard treatment still has limitations in that it can extend a patient’s survival for only about 12 months [7] and that it does not seem to be able to eliminate cancer stem cells (CSCs), as evidenced by frequent cases of tumor relapse and resistance after therapy [8]. Thus, there is an urgent need to identify other novel targets that could be explored for treatment options to improve the therapeutic index. Since HCCs have shown certain common traits selected via genetic and epigenetic alterations [9], targeting any of the common dysregulated genes might provide a new HCC treatment [10].

The disruption of cell cycle regulation is a hallmark for tumor development [11]. Cell division cycle associated 8 (CDCA8) is a component of a chromosomal passenger complex (CPC), which is composed of CDCA8, survivin, INCENP, and Aurora B, and is essential for the stability of the bipolar mitotic spindle. The expression of CDCA8 is widely upregulated in various types of tumors and is required for tumor cell growth and progression [12,13,14,15]. In this study, we found through big data analysis that the overexpression of CDCA8 is significantly correlated with the poor survival of HCC patients. Despite its altered expression, the functions of CDCA8 during HCC progression remain to be determined.

It is well known that a cancer stem cell (CSC) subset is responsible for tumor initiation, migration, recurrence, chemo-resistance, and radio-resistance. We previously reported that upregulation of the CSC marker CD133 promotes HCC development [16]. Identification and characterization of the functional pathways or biomarkers associated with CSC biology will provide useful information for developing novel treatment strategies against both tumor invasion and recurrence.

The purpose of this study is to explore and identify the novel roles of CDCA8 in liver tumor progression. To address this, we first observed whether the inhibition of CDCA8 expression using target-specific siRNA could change HCC phenotypes in vitro and in vivo. We also examined the molecular mechanisms underlying the therapeutic response induced by CDCA8 silencing. We found that targeting CDCA8 inhibits HCC growth and stemness via the restoration of the ATF tumor suppressor and inactivation of the AKT/β–catenin signaling axis. Thus, targeting CDCA8 could be a next-line molecular therapy to effectively reduce the tumor burden and block metastasis or recurrence by eliminating both cancer cells and cancer stem cells in the HCC microenvironment.

## 2. Results

### 2.1. Big Data Analysis of CDCA8 Expression and Its Prognostic Value in HCC

We utilized the publicly available HCC dataset in TCGA with a wider cohort to confirm the relationship between CDCA8 expression and its prognostic value. The level of CDCA8 mRNA in HCC tissues was significantly higher than that in paired, noncancerous tissues (*p* < 0.001) (Figure 1A). Survival analysis showed that higher CDCA8 mRNA levels predicted poorer disease-free survival (*p* = 0.003) and overall survival (*p* < 0.001) of HCC patients (Figure 1B,C).

### 2.2. Silencing of CDCA8 Inhibits HCC Cell Growth, Colony Formation and Migration

Given the statistical relationship between CDCA8 overexpression and poorer patient survival, we first investigated if CDCA8 knockdown could affect HCC cell growth in both the short-term and long-term. To select the siRNA eliciting the highest treatment efficacy and target gene knockdown, we tested three variants of CDCA8-specific siRNAs (CDCA8-1, -2, or -3) at optimal experimental conditions. Huh1 and Huh7 cells were plated at 30% confluence 24 h before transfection and treated with siRNA mixed with the same amount of cationic lipids. Through microscopic observation and cell proliferation assay, we observed that CDCA8-1siRNA and CDCA8-3siRNA caused higher growth suppression than CDCA8-2siRNA in both Huh1 and Huh7 cells after four days of 15 nM treatment compared with cells with negative control (NC)siRNA (Figure 2A,B). Effective degradation of the target mRNA in this condition was affirmed by a RT-PCR (Figure 2C). The extent of cell growth inhibition caused by the three siRNAs was paralleled by a similar degree of target mRNA silencing. In concordance with this short-term observation, the treatment of CDCA8-1siRNA was also effective in blocking long-term colony formation, reducing it by about 70% and 80% in Huh1 and Huh7 cells, respectively, after 12 days of target siRNA treatment (Figure 3A,B). We next investigated the effect of CDCA8 knockdown on the migration ability of HCC cells using a monolayer wound-healing assay. HCC cells treated with CDCA8-1siRNA migrated more slowly than cells treated with a control siRNA (Figure 3C), causing strong inhibition of cell migration of about 50% and 70% in Huh1 and Huh7 cells, respectively, after 96 h of target siRNA treatment (Figure 3D). Similar to the inhibitory effects of CDCA8-1siRNA, the treatment of CDCA8-3siRNA also significantly suppressed the abilities of colony formation and migration in Huh1 and Huh7 cells (Appendix A). This screen identified CDCA8-1 as the most potent siRNA, so it was used for all subsequent studies. We next investigated if CDCA8 overexpression could reverse the phenotypic changes. As expected, a Huh1 transfectant with CDCA8-expressing vector (Appendix A) exhibited an increase in its abilities of proliferation (Figure 4A,B), colony formation (Figure 4C,D) and invasion (Figure 4E,F) when compared to control vector-transfected cells. We then confirmed whether the inhibition of HCC cell growth and migration by CDCA8 knockdown were elicited out by a delay of cell cycle progression. Flow cytometry revealed that, compared to the cells with NCsiRNA, CDCA8 knockdown cells showed an increased percentage of G2/M and decreased percentage of G0/G1 phase cells in both the examined HCC cell lines, suggesting cell cycle arrest in the G2/M phase (Appendix A). This was consistent with previous evidence that CDCA8 promotes G2/M phase transition in cell division [12,17]. These results suggest that CDCA8 is functionally involved in liver tumor cell survival and migration.

### 2.3. CDCA8 Knockdown Upregulates ATF3 and GADD34 Tumor Suppressors to Exert Apoptotic Progression

To find the molecular mechanisms by which CDCA8 silencing could induce the observed phenotypic changes, we performed RNA sequencing to compare the patterns of global gene expression in CDCA8-deficient Huh1 and Huh7 cells to those of NCsiRNA-treated cells. When defined by at least a 2-fold change (*p* < 0.001), treatment of CDCA8-1siRNA disturbed the expression of 145 RNA transcripts in Huh1 and 494 transcripts in Huh7 cells (Figure 5A). Overlapping these two gene sets generated a commonly dysregulated list of 50 genes (25 upregulated and 25 downregulated) (Figure 5A,B). Ingenuity pathway analysis (IPA) showed that the 50 genes were functionally enriched in the top five networks. Notably, as shown in Figure 5C (top network 1) and Figure 5D (top network 2), the antiproliferative effects were driven by the upregulation of tumor suppressive ATF3 [18] and PPP1R15A (GADD34) [19], whereas PCNA and BGLAP, key regulators of proliferation and invasion, were repressed. Western blotting proved that targeting CDCA8 decreases the levels of cyclin B1 and p-cdc2, which mediate the transition of G2/M phase and increase the level of CDKN2B (p15) in both Huh1 and Huh7 cells (Figure 6A). Consistent with transcriptional profiling of the knockdown of CDCA8 expression, the protein levels of the ATF3 and GADD34 tumor suppressors were restored in the same condition. These associations were confirmed in the TCGA data. CDCA8 mRNA expression was negatively correlated with ATF3 (*r* = −0.475, *p* < 0.001) (Figure 6B) and GADD34 (*r* = −0.267, *p* < 0.001) (Figure 6C). Furthermore, ATF3 and GADD34 showed positive associations in HCC (*r* = 0.612, *p* < 0.001) (Figure 6D). The molecular changes upstream elicited an increase in the levels of proapoptotic Bax, cleaved caspase-9, and cleaved PARP-1 (Figure 6A), accelerating apoptotic progression. These findings suggest that restoration of the levels of ATF3 and GADD34 tumor suppressors is a key and common mechanism to incur growth inhibition and apoptotic induction in CDCA8-deficient HCC cells.

### 2.4. CDCA8 Knockdown Inactivates AKT/β–Catenin Signaling and Reduces Stemness

It was previously shown that GADD34 induces apoptosis through inactivation of Akt signaling [20], which contributes to tumor cell migration and invasion [21]. We observed here that the upregulation of GADD34 caused by CDCA8-1siRNA treatment induces a decrease in Akt activity (Figure 7A). GSK-3β is a downstream target of Akt. p-Akt phosphorylates GSK-3β at serine 9 and blocks β-catenin degradation to maintain CSC properties [22,23]. CDCA8 knockdown decreased the levels of p-GSK-3β and β-catenin in both examined HCC cells (Figure 7A). Given the functional significance of Akt signaling in CSC biology and the observation of Akt repression under the CDCA8 knockdown condition, we next chose to determine whether CDCA8 targeting could inhibit stemness in HCC. As expected, the treatment of CDCA8-1siRNA effectively suppressed sphere formation in both Huh1 and Huh7 cells compared to the control siRNA knockdown (Figure 7B). Next, on the basis of our previous observation that the upregulation of CSC marker CD133 contributes to liver tumor initiation [16], we investigated whether the expression of CD133 is also affected by CDCA8 silencing in the CD133^+^ HCC cell population. To accomplish this, the PLC/PRF/5 cell line was chosen, as it was previously reported that the PLC/PRF/5 cell line contains the highest levels of the CD133^+^ cell population among generally available HCC cell lines [24]. First, it was demonstrated that the silencing of CDCA8 decreases mRNA expression of CSC marker CD133 in the PLC/PRF/5 parental cell culture (Appendix A). The CD133^+^ CSC population was then sorted from the CD133^−^ population of PLC/PRF/5 cells by FACS using an APC-conjugated CD133 antibody (Figure 8A,B). The levels of CDCA8 and CD133 in the CD133^+^ CSC portion were significantly higher than in the CD133^−^ portion (Figure 8C). CDCA8 knockdown decreased CD133 expression and increased the levels of ATF3 and GADD34 in the CD133^+^ CSC population (Figure 8D,E). Under the same condition, the AKT/β–catenin signaling axis was also inactivated (Figure 8E). These molecular changes reflected the inhibition of growth and spheroid formation in the CD133^+^ cell population (Figure 8F,G). These data imply the possibility that CDCA8 is functionally associated with cancer stemness, and that CDCA8 knockdown could block the migratory and invasive properties of primary HCC by eliminating CSCs in the tumor microenvironment.

### 2.5. Targeting CDCA8 Suppresses HCC Growth In Vivo

To confirm the suppressive effect of CDCA8 silencing in vivo, a tumor formation assay was performed on BALB/c nude mice. Huh7 cells treated with NCsiRNA or CDCA8-1siRNA for 24 h were subcutaneously inoculated into the left and right flank, respectively, then the size of each tumor was measured for 17 days at 2 days intervals. Seven days after inoculation, there was a clear difference between the control and CDCA8 knockdown groups and the difference consistently increased until 17 days (*p* < 0.01) (Figure 9A). After 17 days, a remarkable difference (*p* < 0.001) was apparent in both the weight and size between the control and CDCA8 knockdown groups, showing about a 5-fold delay in tumor growth kinetics (Figure 9B,C). Next, we investigated if the molecular alterations mediating the inhibition of HCC cell growth and stemness could also be incurred in mice tumors. The method of direct intratumoral injection of CDCA8-1siRNA to a grown tissue was chosen to address this issue. NCsiRNA and CDCA8-1siRNA were mixed with a cationic lipid and injected into tumors three times at 3 days intervals (at Day 0, 3, and 6). Day 0 corresponds to 10 days after inoculations of 5 × 10^6^ Huh7 cells when tumors had reached an average volume of ~50–60 mm^3^. As shown in Figure 9D,E, mRNA and protein levels of CDCA8 were effectively reduced at 24 h and 48 h after the third injection, respectively. Under the same condition, the level of target protein CDCA8 and the number of proliferative cells were significantly reduced in tumors (Figure 9F). The antiproliferative effects were driven by the upregulation of tumor suppressive ATF3 and GADD34, whereas p-Akt and β-catenin, key regulators of stemness, were repressed (Figure 9E). Those changes coincided with observations in parental cancer cells and cancer stem cells.

## 3. Discussion

The upregulation of CDCA8 has been reported in diverse types of cancer, including colorectal, breast, gastric, and lung cancer, implying that inhibition of CDCA8 expression could be an effective therapeutic strategy [14,25,26]. However, the concrete functional roles of CDCA8 in tumor initiation, development, and/or progression are still unclear. Here, we evidenced that siRNA silencing of CDCA8 expression suppresses HCC growth in vitro and in vivo, which demonstrates that CDCA8 is important for the growth of HCC. We next sought to find a molecular explanation of how CDCA8 functions in tumor progression. The changes in cell phenotype were driven by the coordinated and common dysregulation of 50 up- or down-regulated genes, including upregulation of the activating transcription factor-3 (ATF3) tumor suppressor in both mRNA and protein levels [27]. The upregulation of ATF3 was an important molecular response to elicit CDCA8 silencing-mediated HCC growth inhibition. However, the specific regulatory mechanisms of CDCA8 in controlling the expression of ATF3 require further investigation.

In both cancer cells and mice tumors, the increase in ATF3 expression was accompanied with upregulation of GADD34, which is known as a downstream effector of ATF3 and induces apoptosis through inactivation of Akt signaling [20]. In a previous study, it was reported that CDCA8 is functionally involved in mitosis [12]. CDKN2B, the gene encoding the tumor suppressor protein p15in4b, inhibits cell cycle G1 progression. Cyclin B1 and p-cdc2 are known to mediate the transition of the G2/M phase. We confirmed in this study that the silencing of CDCA8 induces the upregulation of CDKN2B (p15) and the downregulation of cyclin B1 and p-cdc2, blocking cell cycle progression. It was also observed with RNA levels that silencing of CDCA8 in HCC cells upregulated the expressions of apoptotic genes, such as CYR61 [28,29] and caspase-7 [30], and another tumor suppressor, KLF2 [31], which was paralleled with the suppression of important genes involved in cell growth (BGLAP [32] and ADAMTSL4 [33]) or cell division (SLBP [34] and cyclin E [35]). Such evidence of molecular alteration may partially explain how targeting CDCA8 could inhibit cell proliferation and induce cell cycle arrest and apoptotic cell death in HCC cells, and suggests the mode of action performed by a CDCA8 inhibitor.

Meanwhile, in terms of the acquisition of stemness in HCC microenvironment, previous evidence indicated that any hepatic lineage cells, such as hepatocytes/cholangiocytes, progenitor cells, and stem cells, can acquire CSC properties through accumulated genetic and epigenetic alterations in diverse signaling pathways, including Akt/WNT/β-Catenin, TGF-β, MET, Hedgehog, MYC, p53, EGF, etc. [36]. Thus, the definition of a target gene functionally associated with CSC biology can provide novel treatment options against HCC to prevent secondary tumor formation. It has been reported by us and other groups that the upregulation of CSC marker CD133 promotes HCC development and the function of CD133 is biologically and clinically implicated in liver CSCs [16,37]. CD133 is a well-known CSC marker which is widely used to isolate CSCs. Further, CD133 contributes to β-catenin-mediated transcriptional activation. Overall, CD133/Akt/GSK3β/β-catenin signaling is one of the key regulatory pathway to maintain CSC properties [23]. The current standard-of-care, sorafenib, a multikinase inhibitor, inhibits tumor angiogenesis and induces apoptosis in HCC by blocking the RAF/MEK/ERK pathway, one of the key functional pathways of HCC development [38]. Previous studies have reported that one of the mechanisms of acquired resistance to sorafenib is mediated by activation of AKT signaling in HCC [8]. We observed in this study that CDCA8 inhibition blocks AKT/GSK3β/β-catenin signaling in both parental HCC cell culture and CD133^+^ cell fraction (Figure 7A and Figure 8E). In particular, the inactivation of the AKT/β–catenin pathway was preceded by the reduction in CD133 expression in CD133^+^ liver CSCs (Figure 8E). These molecular alterations incurred the inhibition of HCC growth and stemness in cellular level. Thus, targeting CDCA8 may contribute to overcoming resistance to sorafenib, and improve the elimination of liver CSCs. We reported here for the first time that the function of CDCA8 is correlated with HCC stemness.

Taken together, our findings suggest the critical role of CDCA8 in HCC growth and stemness, offering key CDCA8-targeting-induced molecular alterations to exert its therapeutic efficacy. The activation of the tumor suppressive ATF3/GADD34 functional pathway and the inactivation of the AKT/β–catenin pathway was observed in general in all the experimental HCC environments of parental cell culture, isolated CSCs, and xenograft-tumors with CDCA8 depletion.

## 4. Materials and Methods

### 4.1. Cell Lines and siRNA Transfection

The human hepatocellular carcinoma cell lines Huh1 and Huh7 were obtained from the Japanese Collection of Research Biosources Cell Bank (JCRB). Both the Huh1 and Huh7 cells were incubated at 37 °C in a humidified incubator with 5% CO_2_ and cultured in Dulbecco’s Modified Eagle’s Medium (HyClone, South Logan, UT, USA) supplemented with 10% fetal bovine serum (HyClone) and 1% penicillin/streptomycin solution (HyClone). The negative control (NC)siRNA, which does not target any endogenous transcripts, served as a control. The NCsiRNA sequences were designed as follows: 5′-ACGUGACACGUUCGGAGAA(UU)-3′(sense) and 5′-UUCUCCGAACGUGUCACGU-3′(antisense). Three variants of CDCA8-specific siRNAs (CDCA8-1siRNA, ID# s30269; CDCA8-2siRNA, ID# 26168; CDCA8-3siRNA, ID# 26260) were purchased from Ambion (Austin, TX, USA). One day prior to transfection, cells were seeded at 30% confluency and treated with 15 nM siRNA by forming a complex with Lipofectamine 2000 (Invitrogen, Carlsbad, CA, USA) in Opti-MEM (Invitrogen).

### 4.2. Cell Proliferation and Clonogenic Assay

For the cell viability assay, we used the CellTiter 96 Aqueous One Solution Cell Proliferation Assay (Promega, Medison, WI, USA) according to the manufacturer’s instructions. For the clonogenic assay, a total of 1 × 10^3^ siRNA transfected cells were seeded in 6-well plates and incubated in the medium for 12 days until the viable cells propagated to sizable colonies for quantification. The colonies formed in each well were fixed with methanol and then stained with 0.5% crystal violet for 30 min. The colonies were counted under a microscope.

### 4.3. Quantitative RT-PCR

RNA was entirely extracted from cells using TRIzol reagent (Ambion Inc., Austin, TX, USA) and synthesized into cDNA using a 1st strand cDNA Synthesis Kit (Takara Biotech, Kusatsu, Shiga, Japan), as recommended by the manufacturer. cDNAs were amplified using a corresponding pair of primers (CDCA8 forward, 5′-GCAGGAGAGCGGATTTACAAC-3′; CDCA8 reverse, 5′-CTGGGCAATACTG TGCCTCTG-3′; GAPDH forward, 5′-GGGAGCCAAAAGGGTCATCATCTC-3′; GAPDH reverse, 5′-CCATGCCAGTGAGCTTCCCGTTC-3′; ATF-3 forward, 5′-GGAGTGCCTGCAGAAAGAGT-3′; ATF-3 reverse, 5′-CCATTCTGAGCCCGGACAAT-3′; GADD34 forward, 5′-CCCAGAAACCCCTAC TCATGATC-3′; GADD34 reverse, 5′-GCCCAGACAGCCAGGAAAT-3′; CD133 forward, 5′-AGTCG GAAACTGGCAGATAGC-3′; CD133 reverse, 5′-GGTAGTGTTGTACTGGGCCAAT-3′) synthesized by Macrogen (Seoul, Korea). The relative quantification of mRNA was performed using a LightCycler 96 (Roche, Basel, Switzerland) according to the manufacturer’s instructions, and was quantified by LightCycler 96 software version 1.1, compared with the Ct (threshold cycle) values of each target gene.

### 4.4. Cell Cycle Analysis

Cells were cultured in 60 mm culture dishes and harvested at 48 h after siRNA transfection. Cells were washed with cold PBS and then fixed for 24 h with 70% cold ethanol at −20 °C. Cells were washed again and cultured for 30 min at room temperature in the dark with in Propidium Iodide (PI) staining solution, with RNase A (BD Biosciences, San Diego, CA, USA). The cell death and cell cycle were analyzed by FACSVerse flow cytometry (BD Bioscience, San Diego, CA, USA) following the manufacturer’s instructions, and quantified using the FlowJo software program.

### 4.5. Wound Healing Assay

For this assay, siRNA-transfected cells were plated in 6-well culture plates with 5 × 10^4^ cells per well and incubated for 24 h to reach the appropriate confluency. The monolayers were scratched with a 200 μL sterile tip, then the floating cells were washed with fresh medium. Cell images were microscopically observed with 40× magnification at five time points (0, 24, 48, 72, and 96 h). Wound closure was analyzed as the ratio of the remaining area of the wound relative to the initial area of the wound, with ImageJ software (National Institutes of Health (NIH), Bethesda, MD, USA). 

### 4.6. Sphere Forming Assay

Cells (5 × 10^3^) were suspended in a serum-free DMEM/F-12 medium containing 4 ng/mL insulin (Invitrogen), 2% B27 (Invitrogen), 20 ng/mL epidermal growth factor, and 10 ng/mL basic fibroblast growth factor, then seeded into 24-well ultralow attachment plates (Corning, Corning, NY, USA). After plating the single cells, cells were transfected with NCsiRNA or CDCA8-1siRNA using Lipofectamine RNAiMAX (Invitrogen) according to the manufacturer’s protocol. The number of cell spheres was counted on day 7 of siRNA transfection.

### 4.7. RNA Sequencing and Functional Network Analysis

The experimental procedures for isolation of total RNA, verification of RNA quality and RNA sequencing were previously described in detail [39]. Sequencing data were deposited at the Gene Expression Omnibus database under accession number GSE77992. By comparing the expression levels of all the human genes in HCC cells treated with control siRNA, the list of genes differentially expressed at least 2-fold in HCC cells with CDCA8 depletion were selected out by using Bootstrap ANOVA method. *p* values of ≤0.001 were considered statistically significant. The significantly overlapping pathways and Gene Ontology categories with differentially expressed genes were analyzed using DAVID (http://david.abcc.ncifcrf.gov, accessed on 7 February 2017) and IPA (ingenuity pathway analysis, www.ingenuity.com, accessed on 10 February 2017).

### 4.8. Magnetic Cell Sorting and Flow Cytometry

Cells were trypsinized and resuspended in PBS with Fcr Blocking Reagent (130-100-857, CD133 MicroBead Kit, Miltenyi Biotec, Auburn, CA, USA). CD133 microbeads were added and incubated for 15 min at 4 °C in the dark. After washes, labeled cells were separated using an autoMACS Pro (Miltenyi Biotec). Magnetic labeled cells and unlabeled cells (CD133+ and CD133- cells) were resuspended in PBS with Isotype Control Antibody, mouse IgG2b, APC (130-122-932, Miltenyi Biotec), or CD133/2 antibody antihuman, APC (130-113-746, Miltenyi Biotec). Cells were incubated for 30 min at 4 °C in the dark. After washes, isolated cells were used to evaluate the efficiency of magnetic separation via FACSCanto II flow cytometry (BD Bioscience, San Jose, CA, USA).

### 4.9. Plasmid Transfection and Overexpression of CDCA8

We first detected the basal level of CDCA8 expression in Huh1 and Huh7 cell lines and observed that the level of CDCA8 expression in Huh1 was lower than in Huh7 (Appendix A). Thus, Huh1 cells were transiently transfected with the pCMV6-entry vector (Empty-vector, PS100001) or CDCA8-expressing vector (CDCA8-vector, RC201385) (all from OriGene, Rockville, MD, USA) using Lipofectamine 2000 (Invitrogen) according to the manufacturer’s protocol.

### 4.10. In Vivo Tumor Growth

Huh7 cells transfected with siRNA for 24 h were harvested with trypsinization and re-suspended in DMEM mixed with Matrigel (Corning, Corning, NY, USA). To explain the efficacy of CDCA8 silencing against in vivo tumors, approximately 2 × 10^7^ Huh7 cells with NCsiRNA or CDCA8-1siRNA were injected subcutaneously into the left and right flanks of four-week-old male BALB/c nude mice (Orientbio, Seongnam, Korea). The tumor size was measured using a Vernier caliper at 2 days intervals for 17 days after tumor injection, and calculated as (the minor axis^2^ × the major axis × 1/2).

To investigate the antitumor effect of CDCA8-1siRNA, an intratumoral injection was performed on BALB/c nude mice. NCsiRNA and CDCA8-1siRNA were mixed with Lipofectamine 2000 (Invitrogen, Carlsbad, CA, USA) and injected directly into tumors three times at 3 days intervals. Day 0 corresponds to 10 days after the inoculation of 5 × 10^6^ Huh7 cells when tumors had reached an average volume of ~50–60 mm^3^. To evaluate the expression of genes in tumor tissues, the tumor tissues were excised at 1 day after injection three times. On day 2 after the injection, the tumor tissues for Western blot analyses were obtained, and the tumor tissues for immunohistochemistry (IHC) were collected 1 week later and used in each experiment.

### 4.11. Western Blot Analysis

After 48 h of siRNA transfection, cells were lysed in RIPA buffer (Thermo Scientific, Rockford, IL, USA) containing 0.01% of protease and phosphatase inhibitor cocktail (Thermo Scientific). The protein concentration was measured with a Pierce BCA Protein Assay Kit (Thermo Scientific, Rockford, IL, USA). The cell protein extracts (50 μg) were separated on SDS-PAGE in 10% gel and transferred to PVDF membranes (Roche, Basel, Switzerland). After blocking with 5% skim milk/Tris-buffered saline plus Tween 20 (TBST), the membranes were incubated with primary antibodies against human ATF-3 (sc-188), GADD34 (sc-8327), p-Cdc2 (sc-12341), PARP-1 (sc-8007), β-actin (sc-4778), GSK-3β (sc-81462) (all from Santa Cruz Biotechnology, Santa Cruz, CA, USA), Cyclin B1 (ab72), Bax (ab32503), Akt (ab126811), CDCA8 (ab70910, Abcam, Cambridge, UK), p-Akt(T308) (#13038), p-GSK-3β(S9) (#9336), β-catenin (#9562), Caspase-9 (#9502), CD133 (#64326, Cell Signaling Technology, Danvers, MA, USA), and Flag (F1804, Sigma Aldrich, Saint Louis, MO, USA). HRP goat antimouse IgG and HRP goat antirabbit IgG (Santa Cruz) were used as the secondary antibodies. The blot signals were visualized with a LAS-3000 Imager (Fujifilm Corporation, Tokyo, Japan). All the uncropped Western blots are shown in Appendix A.

### 4.12. Immunohistochemistry

The tumor of each mouse was fixed with 10% formalin, embedded in paraffin, and sectioned. The sections were stained with hematoxylin and eosin (H&E) or IHC for light microscopic examination. For the assessment of CDCA8 staining, 4 μm sections were permeabilized in PBS and incubated in 10 mM sodium citrate buffer for 20 min at 100 °C with rabbit polyclonal anti-CDCA8 (1:10, Abcam, UK). Sections were then incubated with the secondary antibody (1:200, Santa Cruz Biotechnology, Dallas, TX, USA), visualized using diaminobenzidine chromogen (Vector Laboratories, Burlingame, CA, USA), and counterstained with hematoxylin (Dako, Denmark). The stained section images were microscopically observed with 400× magnification.

### 4.13. Statistical Analysis

Each experiment was repeated at least three times, and the results are presented as the mean ± Standard Error (SE), unless otherwise indicated. The statistical analysis was performed using the Bootstrap *t*-test, Poisson generalized linear model, Student’s *t*-test with equal variance, normality assumption test, or bootstrap ANOVA with contrast tests. The correlations between CDCA8 and other mRNA expressions were assessed with Pearson’s correlation coefficient analysis. Survival curves were elucidated by the univariate Kaplan–Meier estimators and analyzed by the log-rank test. Overall survival was calculated from the date of diagnosis and mortality. Disease-free survival was defined as the time between diagnosis and disease recurrence, or the development of distant metastasis. *p* values of ≤0.05 (*), ≤0.01 (**), and ≤0.001 (***) were considered statistically significant.

## 5. Conclusions

In this study, at the molecular level, we found that silencing of CDCA8 suppresses the growth and stemness of HCC tumor cells through the restoration of tumor suppressor ATF3 and inactivation of the AKT/β–catenin signaling axis. In conclusion, CDCA8 is functionally involved in HCC survival and CSC-like phenotype maintenance, suggesting that molecular targeting of CDCA8 may be an effective systemic therapy for the prevention of tumor recurrence via the elimination of both cancer cells and cancer stem cells in the liver tumor microenvironment.

## Figures and Tables

**Figure 1 cancers-13-01055-f001:**
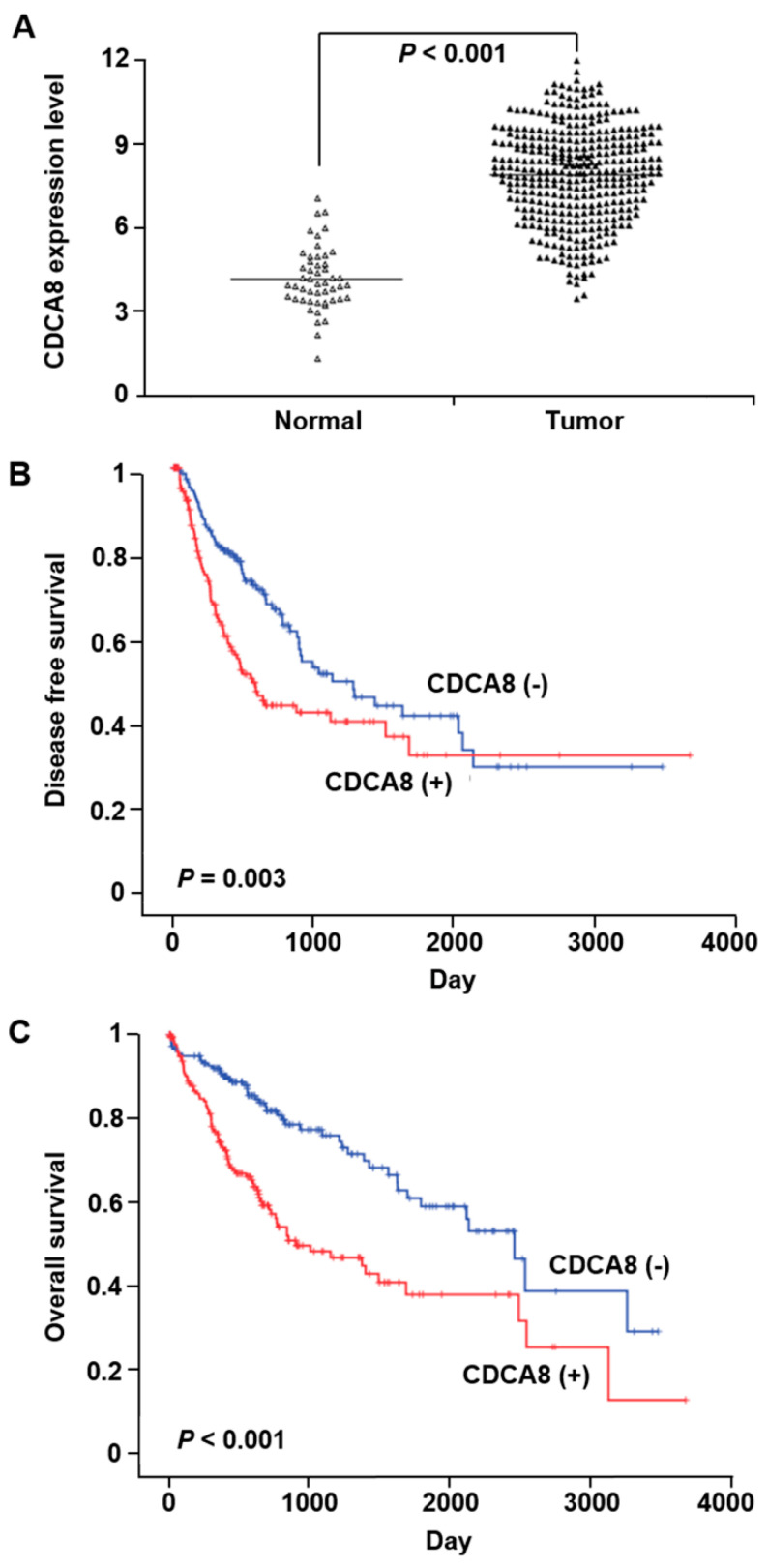
The correlation between CDCA8 overexpression and prognosis in hepatocellular carcinoma (HCC) patients. (**A**) Upregulation of CDCA8 in HCC tissues compared to that in adjacent noncancerous tissues (*p* < 0.001). (**B**,**C**) Associations between CDCA8 expression and the patient’s disease-free survival (*p =* 0.003) (**B**) and overall survival (*p* < 0.001) (**C**).

**Figure 2 cancers-13-01055-f002:**
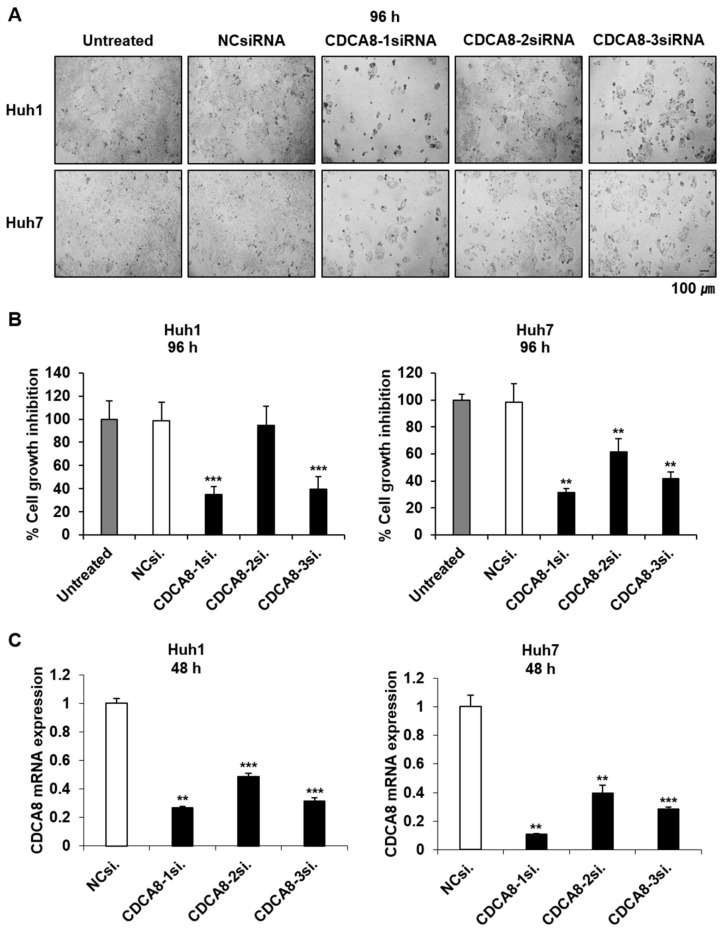
Silencing of CDCA8 suppresses HCC cell proliferation. (**A**) Light microscopy images of Huh1 and Huh7 cells 4 days after NCsiRNA or three kinds of CDCA8-specific siRNA transfection. Scale bar, 100 μm. (**B**) Growth inhibition of Huh1 and Huh7 cells treated with 15 nM siRNA for 4 days. (**C**) Detection of CDCA8 mRNA expression in Huh1 and Huh7 cells at 48 h after transfection. The data are shown relative to GAPDH expression and normalized to NCsiRNA treatment. *NCsiRNA*, negative control siRNA; *CDCA8-1si.*, *CDCA8-2si.*, *CDCA8-3si.*, three different sequences of CDCA8-specific siRNA. All the data in this figure were statistically analyzed using the Bootstrap *t*-test. ** *p* < 0.01; *** *p* < 0.001 vs. control.

**Figure 3 cancers-13-01055-f003:**
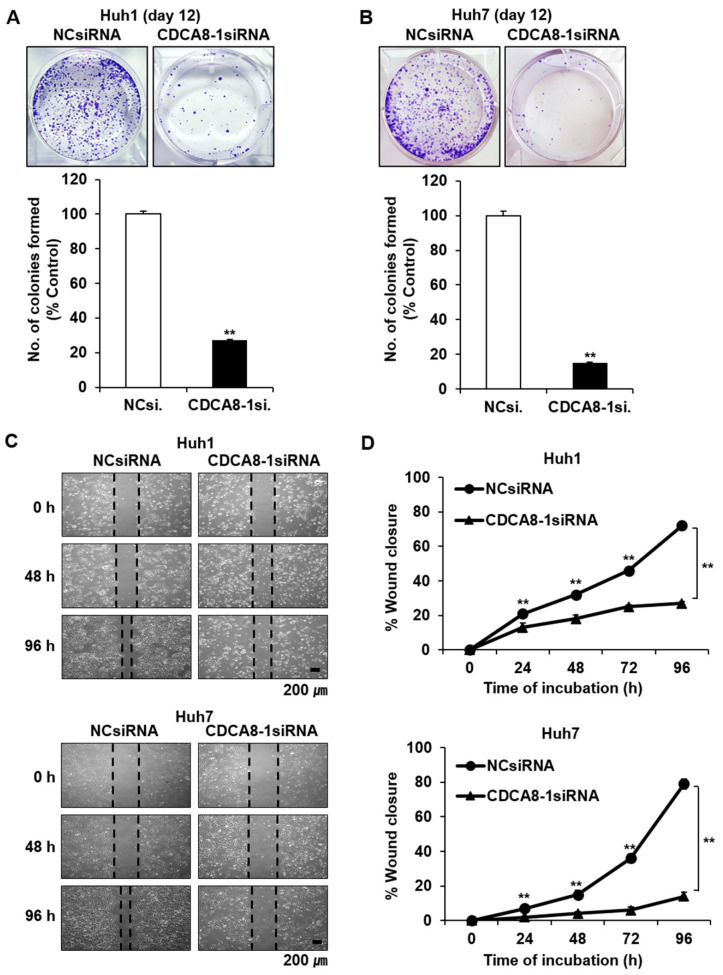
Treatment of CDCA8-1siRNA decreases clonogenicity and migration ability in HCC cells. (**A**,**B**) Observation of changes in the clonogenicity of Huh1 (**A**) or Huh7 cells (**B**) after 12 days of siRNA treatment. The number of colonies was also measured in each cell line (** *p* < 0.01, by the Poisson generalized linear model). (**C**) Observation of HCC cell migration using a wound healing assay. Representative images were taken after 0, 48, and 96 h of siRNA treatment. Scale bars, 200 μm. (**D**) Wound closure was measured daily for up to 4 days and normalized to the time point of day 0 (** *p* < 0.01 by Bootstrap *t*-test, *n* = 3).

**Figure 4 cancers-13-01055-f004:**
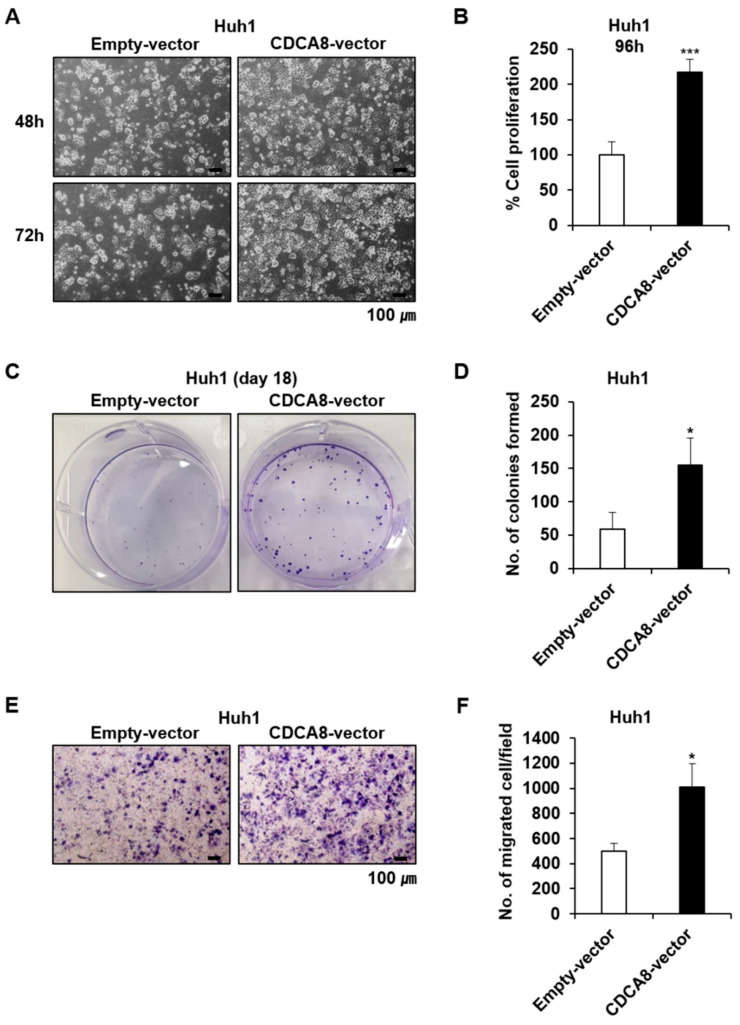
CDCA8 promotes proliferation and migration of CRC cells. (**A**,**B**) Representative light microscopy images (**A**) and the number of proliferative Huh1 cells (**B**) transfected with empty control vector or CDCA8-overexpressing vector for 96 h. (**C**,**D**) Observation of changes in the clonogenicity (**C**) and the number of colonies (**D**) after 18 days of the indicated treatments. (**E**,**F**) Representative light microscopy images (**E**) and the number of migratory Huh1 cells (**F**) transfected with empty control vector or CDCA8-overexpressing vector. Empty-vector, transfected with empty control vector; CDCA8-vector, transfected with CDCA8-expressing vector. * *p* < 0.05; *** *p* < 0.001 vs. control.

**Figure 5 cancers-13-01055-f005:**
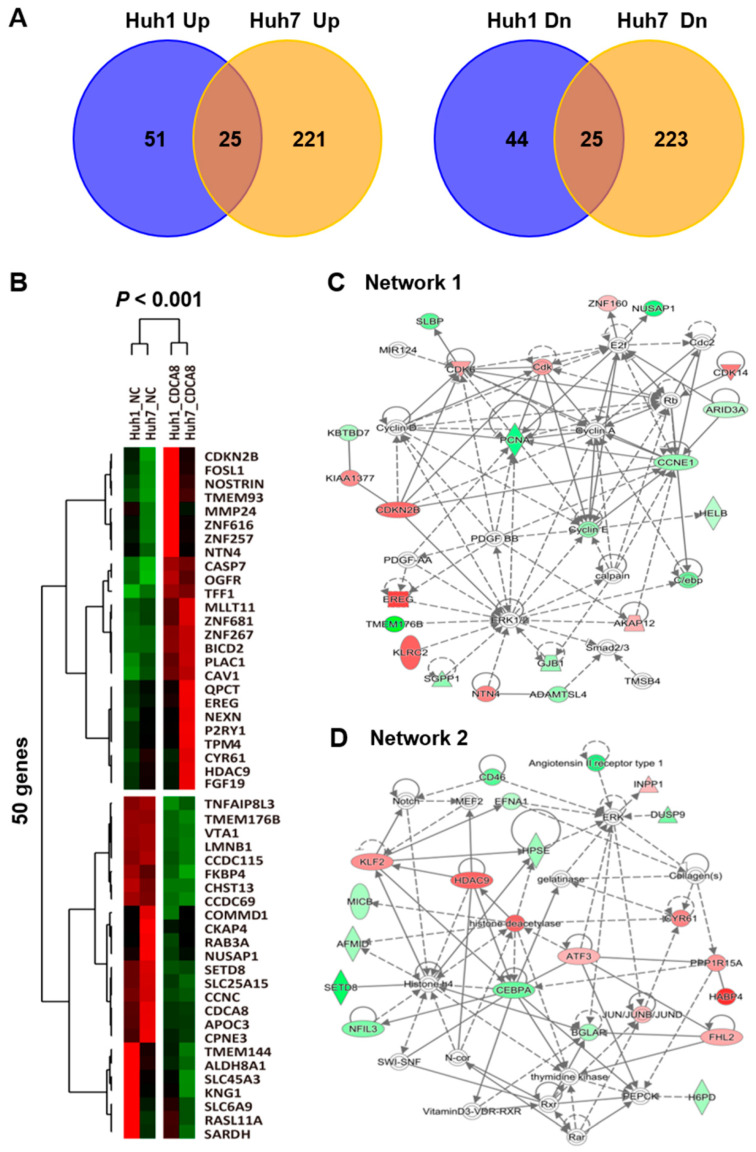
Global observation of dysregulated genes in HCC cells with CDCA8 depletion. (**A**) The number of genes dysregulated in Huh1 and/or Huh7 cells over 48 h of CDCA8-1siRNA treatment. (**B**) A heat map of the 50 commonly up- or downregulated genes in Huh1 and Huh7 cells. *p* < 0.001, red (upregulated) and green (downregulated). (**C**,**D**) Two putative ingenuity pathway analysis (IPA) top networks highly associated with PCNA (**C**) or ATF3 (**D**).

**Figure 6 cancers-13-01055-f006:**
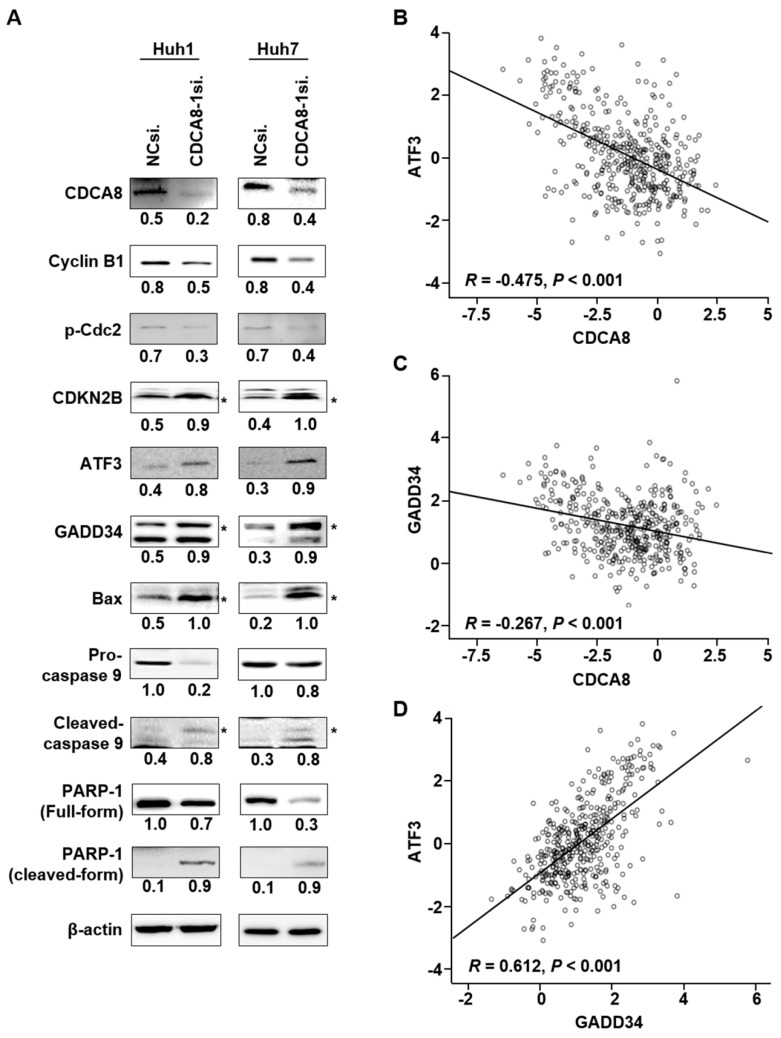
Targeting CDCA8 enhances apoptotic signaling through activation of the ATF3 and GADD34 tumor suppressors. (**A**) Western blot analysis of the expression of CDCA8 and the indicated proteins functionally related to cell proliferation, apoptosis, or cell cycle regulation. The proteins were prepared at 48 h after transfection with CDCA8-1siRNA. β-actin was detected as a loading control. * exact position of target protein band when measured with molecular weight markers (refer to Appendix A). (**B**–**D**) The correlation among the expressions of CDCA8, ATF3, and GADD34 in HCC tissues. TCGA data showed that CDCA8 had a negative correlation with ATF3 (**B**) and GADD34 (**C**), and there was a positive association between ATF3 and GADD34 (**D**).

**Figure 7 cancers-13-01055-f007:**
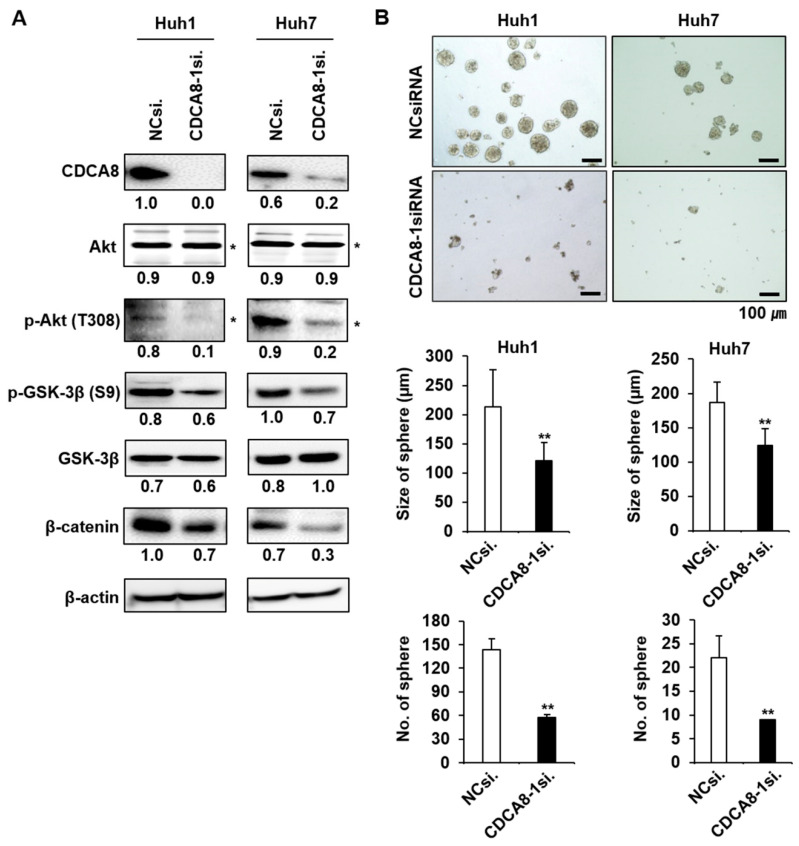
Silencing of CDCA8 inactivates AKT/β–catenin signaling and inhibits sphere formation. (**A**) Western blot analysis of CDCA8 and the indicated proteins functionally involved in AKT/β–catenin signaling. (**B**) Detection of sphere formation in HCC cells treated with NCsiRNA or CDCA8-1siRNA. * exact position of target protein band when measured with molecular weight markers (refer to Appendix A). ** *p* < 0.01 vs. control. Scale bar = 100 μm.

**Figure 8 cancers-13-01055-f008:**
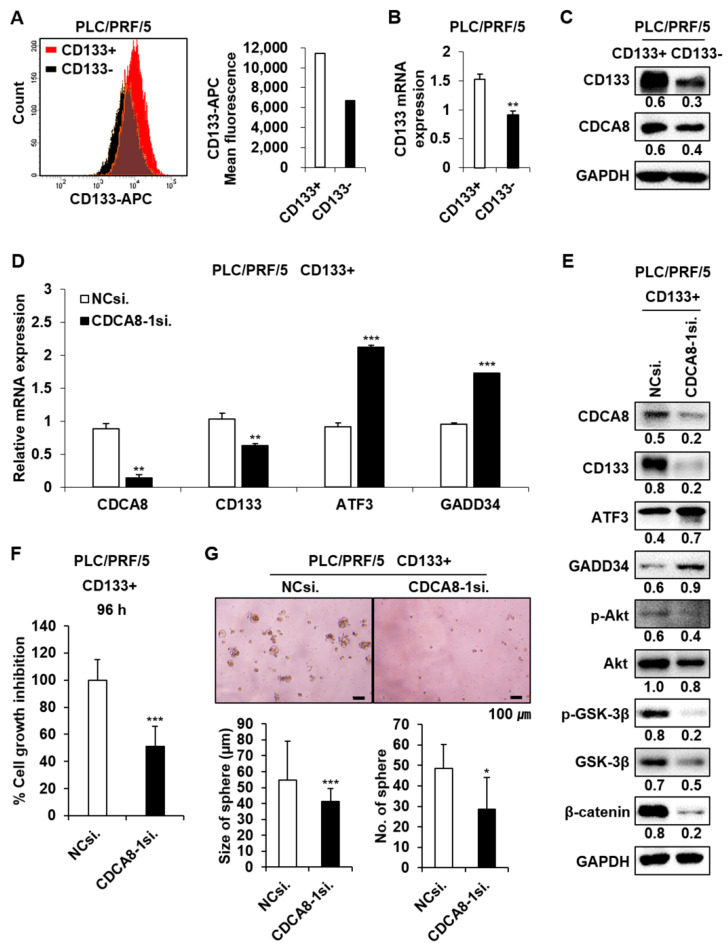
Targeting CDCA8 reduces CD133 expression and inhibits growth and spheroid formation in the CD133^+^ PLC/PRF/5 cell population. (**A**) Isolation of the CD133^+^ population from the PLC/PRF/5 parental cell culture. (**B**) Relative level of CD133 mRNA between CD133^+^ and CD133^−^ cell population. (**C**) Relative levels of CD133 and CDCA8 proteins between CD133^+^ and CD133^−^ cell populations. (**D**) Changes in mRNA expression of the indicated genes in the CD133^+^ population treated with CDCA8-1siRNA. (**E**) Changes in the levels of the indicated proteins in the CD133^+^ population treated with CDCA8-1siRNA. (**F**,**G**) Inhibition of growth (**F**) and spheroid formation (**G**) in the CD133^+^ population treated with CDCA8-1siRNA. * *p* < 0.05; ** *p* < 0.01; *** *p* < 0.001 vs. control. Scale bar = 100 μm.

**Figure 9 cancers-13-01055-f009:**
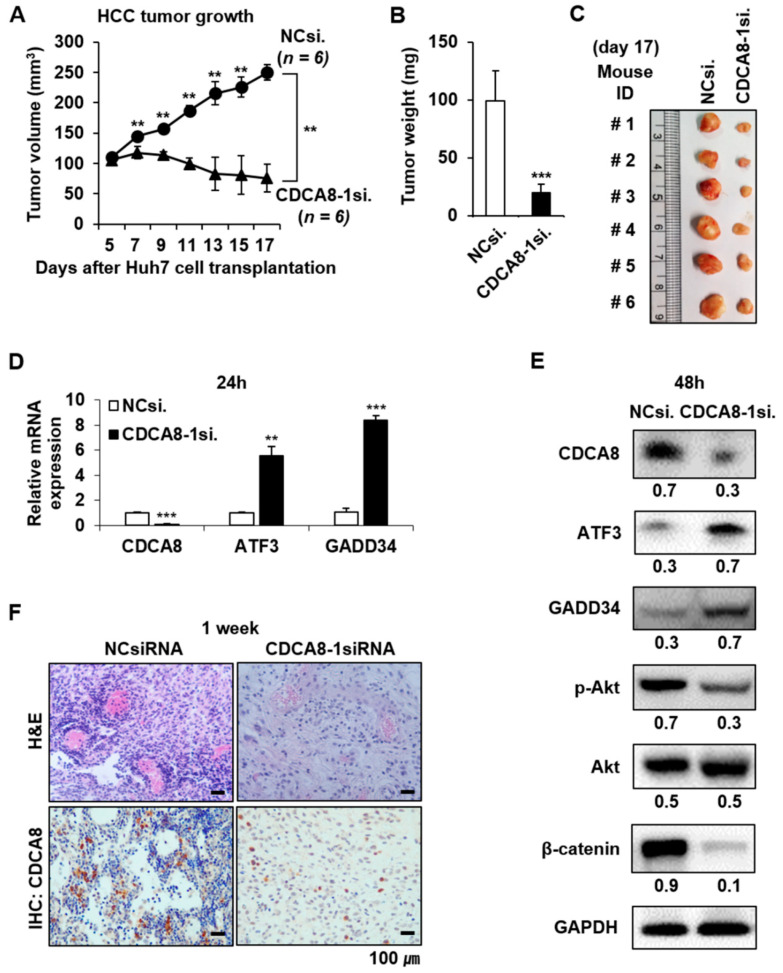
CDCA8 silencing suppresses HCC growth in vivo. (**A**) Kinetics of tumor growth: Huh7 cells transfected with NCsiRNA or CDCA8-1 siRNA for 24 h were subcutaneously injected into nude mice. Tumor diameters were measured on the indicated days with digital calipers (** *p* < 0.01 (*n* = 6 vs. *n* = 6)) by normality assumption test. (**B**,**C**) Average tumor weight (**B**) and gross tumor morphology (**C**) at 17 days after Huh7 cell transplantation. (**D**) Detection of mRNA expression of CDCA8, ATF3, and GADD34 genes in tumors 24 h after the third injection of CDCA8-1siRNA. ** *p* < 0.01; *** *p* < 0.001 vs. control. (**E**) Levels of CDCA8 and the indicated proteins in tumors 48 h after the third injection of CDCA8-1siRNA. (**F**) Detection of proliferative tumor cells and CDCA8 target protein in tumors 1 week after the third injection of CDCA8-1siRNA. Scale bar = 100 μm.

## Data Availability

The data presented in this study are available from the corresponding author on request. The raw data of RNA sequencing were deposited at the Gene Expression Omnibus (GEO) database (http://www.ncbi.nlm.nih.gov/geo, accessed on 14 February 2017) under accession number GSE77992.

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
