# Peer review of "Silencing CDCA8 Suppresses Hepatocellular Carcinoma Growth and Stemness via Restoration of ATF3 Tumor Suppressor and Inactivation of AKT/β–Catenin Signaling"

_cancers, 2021, doi:10.3390/cancers13051055_

Round 1
Reviewer 1 Report
The authors adequately responded to my comments.
Author Response
Thank you very much for your positive evaluation to our revised manuscript. Totally thanks to your constructive comments or questions, we were able to increase the quality of paper and strenghten evidences.
Reviewer 2 Report
This manuscript is a re-submission for publication. The authors provide additional information and do a good job of answering many of the reviewer's critiques of their initial submission. This report is improved and will meet the criteria of this journal. But supplementary figure should be revised because some numbers and content do not match.
Author Response
Thank you very much for your positive evaluation to our revised manuscript. Totally thanks to your constructive comments or questions, we were able to increase the quality of paper and strenghten evidences. I have checked thoroughly about conformity among Supplementary Figures, their legends and the corresponding texts in description of Supplementary Materials (in manuscript), and corrected accordingly only in the revised Supplementary Materials File. The changes are shown in yellow color.
Reviewer 3 Report
In the revised manuscript, the authors responded to reviewer 3's suggestion only partially. By employing more variety of siRNAs and overexpression system, they consolidated their results. They more clearly demonstrated that CDCA8 depletion induced growth suppression and apoptosis. However, the entire work is still highly descriptive. This takes place because at nowhere they proved the role of ATF3 and GADD34 in suppression of malignant phenotypes. They did not attempt to rescue growth suppression by simultaneously depleting these molecules in CDCA8-depleted cells. They did not address even whether overexpression of these genes suppresses cell growth. They are known of tumor suppressive activities, but the authors need to demonstrate that it is the case in the context tested in this work. Without providing more data relating to the core mechanism of growth suppression induced by CDCA8 depletion, this work is still immature to be published from Cancers.
Author Response
Please consider that we have focused in this study to observe the underlying molecular mechanisms mediating CDCA8 silencing-induced inhibition of HCC growth and stemness. We further confirmed through revision (also coping with your constructive suggestions) that the activation of tumor suppressive ATF3/GADD34 functional pathway and the inactivation of AKT/β-catenin pathway was generally induced in all the experimental environments of parental cancer cell culture, isolated CD133+ CSCs, and xenograft-mice tumors, in which were received with CDCA8siRNA and growth suppressive in phenotypes. Using double knockdown or overexpression approach should be helpful to augment the conclusion. However, I thought that, for example, even the comparison of single gene knockdown with double gene knockdown will not be fully enough to elucidate the fundamental question raised by the Reviewer how the function of CDCA8 affects the expression of transcription factor ATF3. That’s why I presented the new data in the revised manuscript without rescue experiments. We did not intend at all to skip over your pinpoint suggestion. We were thinking over and over again how we could resolve this matter. However, at this time, we had a limitation to present further experimental evidences to fully support it as not so many previous clues were available. We were able to find recent evidences that CDCA8 expression may affect the cell cycle and p53 signaling pathways, and overexpression of CDCA8 is significantly related to the mutations of RB1 and p53 (Gao X et al Peer J 2020, 8: e9078). Further, Kuo et al reported that treatment of cisplatin upregulates transcription factors, including ATF3, in p53-mutated subclone variants of hepatoma cells (Oncol Lett. 2016, 12(5): 3723–3730). So, I think, there is a higher probability that the downstream effect of ATF3 upregulation in the condition of CDCA8 knockdown might be involved with p53 biology in HCC. As written down in Discussion briefly (line numbers of 291 through 294), the exact mechanism and functional connection of CDCA8 to ATF3 expression is worthwhile to be further explored in future studies by us or by Cancers readers.
Round 2
Reviewer 3 Report
N/A
This manuscript is a resubmission of an earlier submission. The following is a list of the peer review reports and author responses from that submission.
Round 1
Reviewer 1 Report
In this manuscript, the authors investigated the effect of CDCA8 knockdown in hepatocellular carcinoma cell (HCC) lines. Firstly, they demonstrated using the TGCA cohort that CDCA8 expression is upregulated in HCC tumor tissues, and is positively associated with poor prognosis of HCC patients. In addition, CDCA8 knockdown resulted in the suppression of HCC cell proliferation in vitro and in vivo. This growth suppression was accompanied with reduced migration ability, G2/M arrest, and increased apoptotic cell death. Finally, they further performed transcription profiling in CDCA8-knocked down cells, and demonstrated that ATF3-GADD34-AKT-GSK3β pathway might be involved in the anti-proliferative effects of CDCA8.The findings are interesting, and the data are clear, but preliminary. There is still need to address several issues as follows:
1. Only one siRNA was used throughout this study. This is particularly problematic. The same experiments using additional siRNA targeting CDCA8, or overexpression of CDCA8 should be performed to rule out the possibility that these findings are derived from off-target effects.
2. Figure 2B: Although cell viability is interesting, cell growth is more informative because it tells us whether the number of cells was decreased, or the growth speed was slowed down.
3. Figure 5A: The area Q2 should not be included for the analysis of apoptotic cell death, because it contains necrotic cell death. Only tiny population of sub-G1 (Figure 4A) and Q3 (Figure 5A) could be seen, but this was not consistent with cleaved CCK18 (Figure 5B), CAS9 and PARP (Figure 8A). This point should be rationalized.
4. Although the authors demonstrated that the knockdown of CDCA8 resulted in reduced spheroid formation, this is insufficient to conclude that CDCA8 is involved in cancer stemness. At least, they should investigate 1) CDCA8 expression in cancer stem cells, and 2) cancer stem cell marker expression in CDCA8 knocked down cells.
5. The authors should demonstrate that the downregulation of β-catenin is attributable to the cell growth suppression of CDCA8 knockdown.
6. Discussion is short. Please provides mechanistic insights into the function of CDCA8 by citing more literatures. For instance, hy and how does the spindle-related protein affects ATF3 transcription?
Reviewer 2 Report
In this study, the authors evaluated the function of CDCA8 using HCC cells in detail. The suppression clearly showed the growth inhibition effects. This report is intriguing. The current study does not meet the publishing criteria in this journal and maybe accepted upon revision. I raised several points to improve the content of the report.
Major query
In the result section, the ref16 maybe not necessary because the authors used TCGA data for the analysis. Moreover, I could not validate the prognostic value of CDCA8 as main data in ref16, as mentioned in the introduction section. Was the CDCA8 included in the paper's analysis?
Suppression of CDCA8 in xenografted tumors should be validated at the last observation day because the injected siRNAs in the tumors might not be effective. The expression levels of CDCA8 should be validated in RT-PCR and WB.
In figure 2B, the viability of untreated cells might be suppressed significantly compared to the NCsi cells. The author should show the absorbance as Y-axis in each group. Moreover, the transfection condition of siRNAs in this study may inhibit the cell viability, not depending on the gene suppression effect of CDCA8.
Alteration of AKT/b-catenin, ATF, and GADD34 in xenografted tumors should be validated according to the RNA seq data.
The cancer cells may be eradicated in the CDCA8 suppressed xenograft tumors. HE staining data should be shown.
If authors want to emphasize the relation of CDCA8 and cancer stemness, it should be validated in RNA seq data. Moreover, the definition of picked-up genes should be described in detail in the materials and method section.
Minor query
In the introduction section, the authors described the result data. It may be not necessary for the introduction.
The WB of full protein and cleaved proteins should be shown as one image of the same gels.
Reviewer 3 Report
I conclude this manuscript is too poor to be published from Cancers. These authors argued that ATF3 induction could be the mechanism for growth suppression by CDCA8 knock-down but without performing a rescue experiment using double knock-down. Experiments are overall sub-optimally done, as, for example, Huh1 cell death with control siRNA is huge and there is only insignificant elevation by CDCA8 siRNA (Figure 5). In Figure 6, no histological examination was done. Employing only one siRNA could be dangerous. I recommend the editor not to accept this manuscript.